# Ligand−Structure Effects on *N*−Heterocyclic Carbene Rhenium Photo− and Electrocatalysts of CO_2_ Reduction

**DOI:** 10.3390/molecules28104149

**Published:** 2023-05-17

**Authors:** Lauren Kearney, Michael P. Brandon, Andrew Coleman, Ann M. Chippindale, František Hartl, Ralte Lalrempuia, Martin Pižl, Mary T. Pryce

**Affiliations:** 1School of Chemical Sciences, Dublin City University, D09 K20V Dublin, Ireland; lauren.kearney25@mail.dcu.ie (L.K.); michael.brandon@dcu.ie (M.P.B.); lalrempuia.ralte@mzu.edu.in (R.L.); 2Department of Chemistry, University of Reading, Whiteknights, Reading RG6 6DX, UK; a.m.chippindale@reading.ac.uk; 3Department of Chemistry, School of Physical Sciences, Mizoram University, Aizawl 796004, India; 4Department of Inorganic Chemistry, University of Chemistry and Technology Prague, Technická 5, 16628 Prague, Czech Republic; martin.pizl@vscht.cz

**Keywords:** photocatalysis, electrocatalysis, CO_2_ reduction, rhenium, *N*−heterocyclic carbene

## Abstract

Three novel rhenium *N*−heterocyclic carbene complexes, [Re]−NHC−1−3 ([Re] = *fac*−Re(CO)_3_Br), were synthesized and characterized using a range of spectroscopic techniques. Photophysical, electrochemical and spectroelectrochemical studies were carried out to probe the properties of these organometallic compounds. Re−NHC−1 and Re−NHC−2 bear a phenanthrene backbone on an imidazole (NHC) ring, coordinating to Re by both the carbene C and a pyridyl group attached to one of the imidazole nitrogen atoms. Re−NHC−2 differs from Re−NHC−1 by replacing N−H with an N−benzyl group as the second substituent on imidazole. The replacement of the phenanthrene backbone in Re−NHC−2 with the larger pyrene gives Re−NHC−3. The two−electron electrochemical reductions of Re−NHC−2 and Re−NHC−3 result in the formation of the five−coordinate anions that are capable of electrocatalytic CO_2_ reduction. These catalysts are formed first at the initial cathodic wave R1, and then, ultimately, via the reduction of Re−Re bound dimer intermediates at the second cathodic wave R2. All three Re−NHC−1−3 complexes are active photocatalysts for the transformation of CO_2_ to CO, with the most photostable complex, Re−NHC−3, being the most effective for this conversion. Re−NHC−1 and Re−NHC−2 afforded modest CO turnover numbers (TONs), following irradiation at 355 nm, but were inactive at the longer irradiation wavelength of 470 nm. In contrast, Re−NHC−3, when photoexcited at 470 nm, yielded the highest TON in this study, but remained inactive at 355 nm. The luminescence spectrum of Re−NHC−3 is red−shifted compared to those of Re−NHC−1 and Re−NHC−2, and previously reported similar [Re]−NHC complexes. This observation, together with TD−DFT calculations, suggests that the nature of the lowest−energy optical excitation for Re−NHC−3 has π→π*(NHC−pyrene) and *d*_π_(Re)→π*(pyridine) (IL/MLCT) character. The stability and superior photocatalytic performance of Re−NHC−3 are attributed to the extended conjugation of the π−electron system, leading to the beneficial modulation of the strongly electron−donating tendency of the NHC group.

## 1. Introduction

Inherent in the Intergovernmental Panel on Climate Change’s goal of limiting global temperature increases to 1.5 °C above the pre−industrial average, is the concept of Negative CO_2_ Emissions [1,2]. This implies that the well−recognized requirement to cut greenhouse gas emissions will have to be complemented by the active capture, storage and utilization of CO_2_ [3]_._ On the latter point, CO_2_ can be reduced to yield a range of value−added products, thereby providing an economic and environmental impetus to this activity [4]. Amongst the possible products are fuels (e.g., methanol and methane), commodity chemicals (e.g., formaldehyde, polymers) and chemical precursors (e.g., syngas and C_1_ or C_2_ building block compounds) [5,6,7].

The principal barrier to the practical realization of this vision relates to the intrinsic thermal stability of the CO_2_ molecule, with the C=O bonds possessing a high dissociation energy of ca. 750 kJ mol^−1^ [8,9]. Several different chemical, thermochemical and biochemical processes have been designed to overcome this energy barrier, and, where powered by renewable energy, electrolytic and photolytic reduction processes can also offer the possibility of efficient and, in principle, emission−free CO_2_ conversion [2,6,10,11,12]. Indeed, the term ‘solar fuels’ has emerged to describe the use of abundant solar radiation to affect the reduction of CO_2_ (coupled with H_2_O oxidation), with the harvested energy stored in the bonds of the reduction products [6,13]. The one−electron reduction of CO_2_ to CO_2_^•−^ is highly thermodynamically unfavorable, with a highly negative standard redox potential, *E^0^* = −1.90 V (vs. SHE, pH 7); however, reactions forming two−electron−reduced products are significantly less energetically demanding, with *E^0^* values for CO− and HCOOH−forming processes lying in the range from −0.64 to −0.52 V [8,14,15]. The relatively low electrode potentials associated with the two−electron processes render them attractive for photo− or electro−synthesis, although suitable catalysts are required to minimize the activation energy (or equivalently overpotential). Accordingly, there has recently been much research interest in the development of both heterogeneous [16,17,18] and homogeneous catalysts [7,19,20,21] for efficient and selective photo− and electrocatalytic CO_2_ reduction.

A series of rhenium tricarbonyl compounds based on the so−called *Lehn catalyst*, [Re^I^(bpy)(CO)_3_X] (bpy = 2,2′−bipyridine, X = halide), have been amongst the most studied homogeneous CO_2_ reduction photocatalysts since the initial report of their catalytic ability in 1983 [19,22,23]. In addition to their high specificity for CO production, this interest arises from their rare ability to photocatalytically reduce CO_2_ without the need for a separate, dedicated photosensitizer (PS) moiety [10,24]. In other words, these compounds can absorb solar−spectrum radiation through the creation of an electron−hole pair, with the resulting triplet excited state undergoing reductive quenching (by a reaction with a sacrificial electron donor (SED)) to yield the singly reduced complex that is the (pre−)catalyst of the CO_2_ reduction [14].

More recently, Delcamp and co−workers have reported that the replacement of one of the pyridine rings in bpy with an *N*−heterocyclic carbene (NHC) group may induce an enhancement in the turnover for CO formation. This is particularly true when an electron−withdrawing moiety, such as a phenyl−CF_3_ group, is a component of the NHC ligand (see Figure 1A), with TONs increasing from 31(for [Re(bpy)(CO)_3_Br]) to 50 (for the NHC−Ph−CF_3_ analogue) under the same conditions [10,25]. *N*−heterocyclic carbene ligands are usually five−membered rings, containing the carbene carbon bound to at least one of the imidazole nitrogen atoms [26,27]. The improvement in the catalytic performance on replacing pyridine with NHC was attributed to the strong σ−donating character of the NHC group, in contrast to the electron deficient pyridine ring π−system, which resulted in a cathodic shift in the first reduction potential. The associated change in the electronic properties of the complex was postulated to beneficially tune the position of the reduction potentials relevant to the electron transfer to CO_2_. To optimize the effect, some counter balancing of the electron density at the NHC ring was found to be useful, as demonstrated by the improved catalytic activity arising from the introduction of an electron−withdrawing group at the phenyl ring attached to the NHC ring [10,26]. Delcamp and co−workers followed up their initial report with a series of systematic studies in which they replaced Br^−^ with other axial ligands, such as Cl^−^ and P(OEt)_3_ [28], and −CF_3_ with other electron−withdrawing groups (such as −NO_2_ or −CN), and additionally investigated the impact of experimental conditions on the photocatalytic performance [10,29]. The utilization of Re−NHC dicarbonyl complexes as photocatalysts for CO_2_ reduction has been reported by other researchers [30,31].

The Delcamp group has demonstrated that various Re−NHC tricarbonyl compounds (including the –phenyl−CF_3_ substituted molecule in Figure 1A) also function as homogeneous electrocatalysts for CO_2_ reduction [10,25,28]. Interestingly, an enhanced current was noted in the presence of dissolved CO_2_ at the first reduction wave of these Re−NHC complexes, in contrast to most Re−based catalysts (including the Lehn catalyst), where the onset of catalysis generally coincides with the second reduction wave [25]. Electrocatalytic CO_2_ reduction at the first reduction wave has also been observed for a few other Re−NHC complexes [30,33] and closely related Re−mesoionic carbenes [34], although there are exceptions [35]. A comprehensive review of Re−NHC photo− and electrocatalysts for CO_2_ reduction has recently been published elsewhere [24].

In this contribution, we report the syntheses and characterization of three novel rhenium−NHC tricarbonyl compounds (see Figure 1B). Both Re−NHC−1 and Re−NHC−2 feature a phenanthrene backbone fused to the NHC ring but differ in the attachment of a benzyl group to one of the NHC nitrogen atoms in Re−NHC−2 instead of N−H in the same position in Re−NHC−1. This design philosophy was pursued to explore the possibility of a proton−assisted CO_2_ reduction process. In Re−NHC−3, the phenanthrene backbone is replaced by a pyrene group, with the aim of extending the π−electron conjugation on the imidazole side of the ligand, thereby shifting the light absorption to longer wavelengths, and facilitating photocatalysis with lower−energy visible light (470 nm). In the case of Re−NHC−1 and Re−NHC−2, photocatalysis only occurred following UV irradiation (355 nm). Cyclic voltammetry measurements were carried out to probe the thermodynamic feasibility of the complexes for photocatalytic CO_2_ reduction, and also to assess this series of Re−NHC compounds for electrocatalytic CO_2_ reduction. The onset of a catalytic current response was evident at the first reduction wave in both Re−NHC−2 and Re−NHC−3, although the activity significantly increased at the potential of the second reductive wave in both cases. A different sequence of reduction reactions prevails in the case of Re−NHC−1, thereby accounting for the more negative onset of electrocatalytic CO_2_ reduction for this complex. Infra−red spectroelectrochemistry was employed to assign the reduction steps observed in the cyclic voltammograms. DFT calculations were also conducted to support the experimental observations.

## 2. Results and Discussion 

### 2.1. Synthesis and Structural Characterization

All three complexes investigated, Re−NHC−1−3 (Figure 1B), were synthesized using similar methods to those previously reported for this class of Re tricarbonyl complexes [29] (see the Materials section in Experimental). All compounds were characterized using ^1^H and ^13^C NMR spectroscopy, including the precursor ligands (see Appendix A), elemental analysis, mass spectrometry and IR spectroscopy. The spectroscopic data are fully consistent with the formulae and facial arrangement of the three carbonyl ligands.

The structural parameters for Re−NHC−1 (Figure 1, Appendix A), determined from single−crystal X-ray diffraction, indicate that the Re(I) metal center displays a slightly distorted octahedral coordination geometry, with three facially oriented carbonyl ligands, the chelating bidentate NHC−pyridine unit and the axial bromide anion. The Re−CO bond distances are similar to those reported for related Re−NHC complexes [36]. The Re−CO bond length trans to the pyridyl group is slightly shorter (1.915(3) Å) than the Re−CO bonds trans to Br (1.940(3) Å) and NHC (1.974(4) Å). The influence exerted by the NHC ring is apparent only when considering the bond length of the whole Re−CO moiety. The DFT−calculated bond lengths and angles are in good agreement with the experimental X-ray diffraction results (Appendix A).

### 2.2. Photophysical Investigations

#### 2.2.1. Electronic Absorption Spectroscopy

Re−NHC−1 and Re−NHC−2 show very similar UV−Vis absorption spectra between 230 and 430 nm (see Appendix A), which are comparable with the spectra previously reported for the related Re−NHC tricarbonyl complexes. For example, Casson et al. [37] reported three absorption maxima at 265, 279 and 358 nm. Li et al. [38] detailed similar findings and assigned the UV absorption (220–280 nm) as intraligand (IL) π→π* transitions within the pyridine−functionalized NHC (pyridine−NHC) ligand and the near−UV absorption (around 360 nm) to a metal−to−ligand charge−transfer (MLCT) transition, *d*_π_(Re)→π*(pyridine−NHC). For Re−NHC−1 and Re−NHC−2, an intense UV band is evident in each electronic absorption spectrum at ca. 250 nm. In addition, there are several shoulders between 270 and 330 nm, a low−intensity band with the maximum absorption at 358 nm and additional shoulders at both sides (see Figure 2a). The accurate assignment of the underlying electronic transitions in Re−NHC−1 has been based on the analysis of the TD−DFT data obtained for a reliable model of the complex. The calculated results are summarized in Appendix A. The strong absorption at 250 nm comprises several close−lying electronic transitions dominated by π(phenanthrene)→π*(pyridine)/π*(NHC−phenanthrene) and π→π*(NHC−phenanthrene), with a contribution from *d*_π_(Re)π*(NHC−phenanthrene) (MLCT). The pronounced shoulder at 290 nm is assigned to *d*_π_(Re)/π(phenanthrene)→π*(pyridine)/π*(NHC−phenanthrene), with a contribution from *dp*(Re−Br)→π*(pyridine). Further, the absorption band at 358 nm is attributed to *dp*(Re−Br)→π*(pyridine). This dominant metal−ligand−to−ligand charge transfer, (M−L)LCT, transition originates in HOMO−1, the frontier occupied photoactive orbital localized on the formally Re(I) center and the axial bromido ligand, and the electronic excitation is directed into empty π*(pyridine−NHC)[37,38,39,40,41]. Finally, the poorly resolved shoulder around 400 nm was also encountered in the reported electronic absorption spectrum of the reference Re−NHC complex depicted in Figure 1 [10], and corresponds to the HOMO→LUMO electronic transition that exhibits a combined *d*_π_(Re)/π(phenanthrene)→π*(pyridine−NHC) (partial MLCT) character. Given the strong similarity of the electronic absorption spectra of Re−NHC−1 and Re−NHC−2, we believe that the latter complex undergoes the same kind of electronic excitations, despite the replacement of the N−H moiety in Re−NHC−1 with the benzyl substituent.

The electronic absorption spectrum of Re−NHC−3 in the near−UV region differs markedly from those of the two other rhenium complexes in the series. In Re−NHC−3, the intense intra−ligand π→π*(pyridine−NHC) absorption band is slightly blue−shifted to 244 nm. Two additional prominent absorption bands arise at 295 and 348 nm. Based on TD−DFT calculations (see Appendix A), the electronic absorption transition at 295 nm is dominated by *d*_π_(Re)/π(pyrene)→π*(pyridine)/π*(NHC−pyrene) (IL/MLCT), with contributions from *dp*(Re−Br)→π*(pyridine/NHC−pyrene) ((M−L)LCT) and π(NHC−pyrene)→π*(pyridine)/π*(NHC−pyrene). The absorption band at 348 nm is assigned to π(NHC−pyrene)→π*(pyridine)/π*(NHC−pyrene), with a contribution from *dp*(Re−Br)→π*(pyridine/NHC−pyrene). The shoulder at ca. 328 nm belongs to π(NHC−pyrene)→π*(pyridine) (LLCT) and π→π*(pyrene). The weak lowest−energy absorption at ca. 400 nm is dominated by the HOMO→LUMO transition having a mixed π→π*(NHC−pyrene) and *d*_π_(Re)→π*(pyridine) (IL/MLCT) character. Finally, the prominent shoulder at ca. 384 nm involves two charge−transfer transitions, *dp*(Re−Br)→π*(pyridine/NHC−pyrene). The assignment of the low−lying absorption features most likely applies also to those observed in Re−NHC−1 and Re−NHC−2. The deviating intense electronic absorption of Re−NHC−3 in the near−UV region (280–360 nm) has its main origin in the ππ* (IL) excitation within the large π−conjugated pyrene system.

#### 2.2.2. Luminescence Spectroscopy

Following excitation, emission spectra for Re−NHC−1−3 were recorded at room temperature in acetonitrile (see Figure 2b), and at 77 K in an ethanol−methanol 4:1 (*v*/*v*) glass (see Appendix A). In solution, following excitation at 375 nm, broad structureless emission bands were observed with maxima at 490 nm and 496 nm for Re−NHC−1 and Re−NHC−2, respectively, while the emission maximum for Re−NHC−3 is notably red−shifted to 527 nm. Such characteristics point to emission from a ^3^MLCT state [38,42], which is consistent with the dominant *dp*(Re−Br)→π*(pyridine/NHC−pyrene) electronic absorption transition calculated for Re−NHC−3 in the near−UV region at the excitation wavelength. Following the excitation of the complexes at 77 K, all three emission spectra became significantly structured. The changes in the spectra at 77 K are likely to be caused by LC ππ* transitions. At room temperature, the compounds are capable of reaching a thermal equilibrium at a faster rate than the emission lifetime, which causes the (M−L)LCT state to be lower in energy and emitting. At 77 K, due to the rigidity in the matrix environment, the compound cannot be thermally equilibrated at a similar rate to the emission lifetime. The energy gap between the MLCT state and the LC state is smaller, so these excited states may eventually invert, causing the emission to originate from the LC state [37,38,42,43,44].

The vast majority of Re^I^−NHC tricarbonyl complexes reported in the literature exhibit blue−shifted luminescence compared with electronically related Re^I^−(α−diimine) tricarbonyl complexes [10,37,38,39,45,46]. For example, Worl et al. and Koike et al. have reported the emission maxima for a range of Re^I^−(α−diimine) tricarbonyl complexes, and all are below 530 nm [45,46]. In the case of the Re−NHC tricarbonyl complexes, the emission maxima are generally found at shorter wavelengths, below 500 nm [10,37,38]. This trend was attributed to the strongly σ−donating nature of the NHC ligands [39]. In this context, the emission maxima we observed for Re−NHC−1 and Re−NHC−2 (490 and 496 nm, respectively) may be regarded as typical for rhenium−NHC tricarbonyl complexes, while the emission of Re−NHC−3 (527 nm) is at a lower energy than expected for a Re−NHC complex. This observation can be rationalized in terms of the increased conjugation introduced by the pyrene backbone in Re−NHC−3, which tends to modulate the electron−donating tendency of the NHC ring. This explanation is supported by the calculated strong participation of the π−π*(pyrene−NHC) system in the lowest−energy excitation of Re−NHC−3 (see the preceding section).

The emission lifetimes for the Re−NHC−1−3 complexes were determined in CD_3_CN solutions. Following excitation at 375 nm, all three complexes exhibited biexponential decays, with the shorter and longer lifetimes in the range of 5−12 ns and 26−941 ns, respectively (Table 1 and Appendix A). These results agree with previous studies of similar Re−NHC tricarbonyl complexes [32,36,37,39]. Nonetheless, there is no discernible trend, and the differences cannot be attributed to either the NHC backbone (pyrene, phenanthrene) or the N−R substituent (R = H, benzyl). Re−NHC−2 displays the longest lifetimes, viz. τ_1_ =11.6 ns, τ_2_ = 940.9 ns, whilst those of Re−NHC−1 and Re−NHC−3 are 10× and almost 40× shorter (for τ_2_), respectively. The lowest−energy electronic absorptions of Re−NHC−1−3 do not differ substantially (see Figure 2). If the pyrene backbone in Re−NHC−3 is responsible for the shortest emission lifetime in the series, then the large difference in the lifetimes between Re−NHC−1 and Re−NHC−2 is not in accordance with the phenanthrene backbone being present in both complexes. The nature of the optically excited states and their conversion to the emissive excited states need to be studied in greater detail with time−resolved laser techniques (TA, TRIR, TR3) to unravel the factors determining the photophysical behavior of Re−NHC−1−3.

#### 2.2.3. Photostability

The exposure of Re−NHC−1 and Re−NHC−2 in acetonitrile to UV light had a more pronounced effect on the luminescence spectra compared to the UV−vis absorption spectra. Following the progressive excitation of Re−NHC−1 with 375 nm photons, the emission band at 492 nm decreased in intensity and a new band grew at 617 nm (see Figure 3). Similarly, for Re−NHC−2, the luminescence band at 497 nm slowly disappeared and a photoproduct started to emit at 650 nm. In contrast, the intensity of the emission spectra of Re−NHC−3 decreased only slightly following 375 nm excitation, and the overall shape of the luminescence profile (Figure 2) was unchanged. The same red shift in the luminescence spectra was observed for both Re−NHC−1 and Re−NHC−2 upon prolonged irradiation with the UV component of either daylight/ceiling light (see Appendix A).

The UV−vis absorption spectra of Re−NHC−1 and Re−NHC−2 (Appendix A) remained unchanged in the dark (over 24 h). Upon exposure to UV light, however, new absorption bands formed for Re−NHC−1 at ca. 300 and 335 nm with a further absorption from ~400 to 480 nm (Appendix A). Similar behavior was observed for Re−NHC−2 (Appendix A). This red shift in the visible absorption spectra was also observed in the luminescence spectra when Re−NHC−1 and Re−NHC−2 were irradiated with UV light. In contrast, the UV−Vis absorption spectrum of Re−NHC−3 remained essentially unchanged, highlighting the difference in photostability within the Re−NHC series controlled solely by varying the NHC backbone from phenanthrene in Re−NHC−1 and Re−NHC−2 to pyrene in Re−NHC−3.

Changes in emission spectra closely resembling those observed for Re−NHC−1 and Re−NHC−2 have been reported in the literature for other Re^I^−NHC tricarbonyl complexes. Casson et al. assigned the growth of a new red−shifted emission band to the displacement of the bromide ligand, which is known to be labile in acetonitrile [37]. Vaughan et al. carried out IR and ^1^H NMR studies to monitor the reactivity triggered by UV irradiation of CD_3_CN solutions containing Re−NHC tricarbonyls (with phenyl and pyridyl substituents on NHC nitrogen atoms and lacking the aromatic NHC backbone).[36] They also proposed the initial loss of the axial halide ligand (Br^−^ or Cl^−^) and its replacement with a solvent molecule (acetonitrile), leading to a cationic product. The cationic solvento complex further underwent the photochemical dissociation of a CO ligand, transforming it to a dicarbonyl complex containing both acetonitrile and chloride ligands [36]. By analogy, the proposed photochemical transformations of the neutral Re−NHC−1 and Re−NHC−2 complexes in acetonitrile to the corresponding solvento cations should stabilize the LUMOs residing on the NHC−1 and NHC−2 ligands (except for the benzyl substituent in Re−NHC−2), similar to the photostable Re−NHC−3 (see Appendix A). This assumption receives strong support from the shift in the reduction potential from −2.01 V (vs. Fc^+^/Fc) for the neutral Re−NHC−2 in acetonitrile to −1.82 V for the cationic photoproduct (see Appendix A).

### 2.3. Redox Behaviour 

#### 2.3.1. Electrochemical Reduction and Electrocatalysis

***Cyclic voltammetry.*** A purpose−driven spectro−electrochemical study was performed to evaluate the thermodynamic feasibility of Re−NHC−1−3 as phototocatalysts for CO_2_ reduction, as well as to probe the capacity of their two−electron−reduced forms to act as electrocatalysts for CO_2_ reduction. 

The nature of the reductive processes for the broader family of rhenium tricarbonyl halide compounds has been extensively studied and described in the literature [47,48,49]. It is generally accepted that the first cathodic wave corresponds to a ligand−based one−electron reduction, initially yielding the radical anion [complex−halide]^•−^, which tends to decompose through halide loss in an EC process [47]. Depending on the nature of the solvent (Sv), the five−coordinate secondary product may either coordinate to a solvent molecule to form the neutral radical [complex−Sv]^•^, or may be subject to dimerization. The parent Lehn catalyst, [Re^I^(bpy)(CO)_3_(halide)] (bpy = 2,2′−bipyridine), exhibits quasi−reversible voltammetry for the first reduction step in acetonitrile, suggesting that the halide loss is relatively slow [25,33]. By contrast, reported Re−NHC tricarbonyl halide complexes display irreversibility at the initial reduction event, indicating that halide loss is more facile compared to that in the bpy analogue [25,33,35,38,50]. On the basis of a combination of voltammetry, IR and electron paramagnetic resonance (EPR) spectroelectrochemistry, together with DFT calculations, it has been suggested by Suntrup et al. [51] that the first reduction of Re−NHC tricarbonyls most likely leads to the formation of more than one species in equilibrium, including [complex−halide]^•−^, the solvent−substituted radical [complex−Sv]^•^ and possibly the five−coordinate radical [complex−5c]^•^, which may be stabilized by the strongly electron−donating NHC ligand.

Cyclic voltammograms (CVs) characterizing the reductive electrochemistry of Re−NHC−1−3 are presented in Figure 4a. Re−NHC−2 and Re−NHC−3 in acetonitrile/Bu_4_NPF_6_ show very similar, totally irreversible two−electron reduction waves at almost identical peak potentials (wave R1, Table 2). This observation is not surprising, given the strong delocalization of the π* LUMO of Re−NHC−3 over the pyridyl−NHC−pyrene rings (Appendix A). The replacement of the pyrene backbone with phenanthrene in Re−NHC−2 apparently has a negligible effect on the reduction potential. The two−electron nature of the initial (ECE) reduction step, R1, for Re−NHC−2 and Re−NHC−3, can be deduced from a comparison with (a) R1 of Re−NHC−1 that will be shown below to belong to a different, one−electron process, and (b) reversible one−electron R3 in the reduction course of Re−NHC−3. For Re−NHC−2 and Re−NHC−3, the five−coordinate radical, [complex−5c]^•^, formed by rapid bromide dissociation from the initially generated [complex−Br]^•−^, does not directly dimerize or coordinate acetonitrile, but instead reduces directly at the cathodic surface to [complex−5c]^−^. The latter two−electron−reduced species will react with a yet non−reduced parent [complex−Br], diffusing from the bulk solution to the cathode to form the Re−Re dimer, [{complex}_2_]. The latter product reduces further at R2 to regenerate the two−electron−reduced species, [complex−5c]^−^. The quasi−reversible nature of R2 may reflect the slow dissociation of the Re−Re bond in the primary single−reduced dimer [{complex}_2_]^•−^. The reversible reduction wave R3 (*E*_1/2_ = −2.49 V vs. Fc^+^/Fc) along the cathodic route of Re−NHC−3 then belongs to the one−electron reduction of the pyrene backbone on NHC−3, generating the three−electron−reduced species, [complex−5c]^2−^. Indeed, free pyrene reduces in acetonitrile at *E*_1/2_ = −2.61 V vs. Fc^+^/Fc [52]. For Re−NHC−2, the reduction of the phenanthrene backbone of NHC−2 in [complex−5c]^−^ is negatively shifted to the onset of the large reduction wave at −2.95 V (see Appendix A), indicating the decomposition of the highly reduced species. The negative potential shift of R3 corresponds to the difference of ca. 250 mV in the reduction potentials of free pyrene and phenanthrene [53].

The cyclic voltammogram of Re−NHC−1 (Figure 4a) indicates a different reduction path than described above for Re−NHC−2 and Re−NHC−3. The initial, totally irreversible reduction wave R1 features a significantly lower peak current, pointing to the one−electron nature of the cathodic process with initial population of the π*(pyridine−NHC) LUMO of the complex (Appendix A). A new close−lying reduction wave R1′ appears on a continued negative potential scan, which is absent in the CV responses of Re−NHC−2 and Re−NHC−3. This behavior can be explained by the rapid cleavage of the N−H bond in the singly reduced NHC−1 ligand and the reductive elimination of the H atom, converting [complex−Br]^•−^, generated at R1 of Re−NHC−1, to [complex′−Br]^−^. The latter anionic species reduces further at negatively shifted R1′ to the corresponding unstable radical dianion, and the Re−Br is expected to cleave at this stage. The ultimate reduction product at the broad wave R2′ is tentatively assigned as [complex′−5c]^2−^, stabilized by the extra negative charge at NHC−1(−H). 

***Electrocatalysis.*** A direct comparison of the cyclic voltammograms recorded for Re−NHC−1−3 in the acetonitrile electrolyte under a dry N_2_ atmosphere and under CO_2_ (see Figure 4b–d) reveals catalytic currents originating mainly at R2 for Re−NHC−2 and Re−NHC−3, and R2′ for Re−NHC−1. These results are not surprising, having revealed that the electrocatalyst is, in all three cases, the two−electron−reduced five−coordinate complex similar to the archetypal [47] catalyst, [Re(CO)_3_(bpy)]^−^, viz. [complex−5c]^−^ for Re−NHC−2 and Re−NHC−3, and probably also [complex′−5c]^2−^ for Re−NHC−1. In the latter case, the electrocatalytic process may also be coupled with a proton transfer due to the reductive cleavage of the N−H bond in the NHC−1 ligand, offering new activation routes while increasing the complexity of the system [54].

***Infra−red Spectroelectrochemistry.*** To support the tentative, although well−argued, assignments of the reduction steps observed in the cyclic voltammograms of Re−NHC−1−3 in the preceding section (Figure 4 and Table 2), infra−red spectroelectrochemical (IR SEC) measurements were conducted with an air−tight, optically transparent, thin−layer electrochemical (OTTLE) cell filled with the electrolyte solutions of the complexes in weakly coordinating THF. The assignment of the reduction products at this stage is mainly based on analogy with the electronically and structurally closely−related Re−bpy and Re−(py−NHC) tricarbonyl complexes [33,47,54]. The course of the reduction path was monitored by thin−layer cyclic voltammetry.

In summary, parent Re−NHC−2 (ν(CO) at 2018, 1924 and 1895 cm^−1^ in THF/Bu_4_NPF_6_), denoted as [complex−Br], reduces at R1 to the dimer [{complex}_2_] (ν(CO) at 1982, 1948, 1874 and 1862 cm^−1^), proving the rapid cleavage of the Re−Br bond in the singly−reduced [complex−Br]^•−^ that was not detectable (Figure 5). At ambient temperature, there was also no IR spectroscopic evidence obtained for the ECE process via the two−electron−reduced five−coordinate [complex−5c]^−^ ion preceding the dimerization step, which is observed in conventional cyclic voltammetry. To recover [complex−5c]^−^ (ν(CO) at 1912, 1811br cm^−1^), the cathodic potential needs to be shifted to R2 (Table 2). The latter complex exhibits the characteristic CO−stretching band pattern with broad bands (reflecting the negative charge) and unresolved ν_2_ and ν_3_ modes (reflecting the dynamic behavior of the Re−tricarbonyl unit in the five−coordinate structure) at a small wavenumber (reflecting the formal Re(0) oxidation state). The experiment required strictly dry and inert conditions to avoid side reactions of the reduced complexes.

In contrast, Re−NHC−1 (ν(CO) at 2017, 1923 and 1892 cm^−1^), which differs from Re−NHC−2 due to the presence of an N−H bond instead of the imidazole−benzyl substituent but shows an almost identical R1 potential (Table 2), reduces at R1 to a mononuclear tricarbonyl product with the ν(CO) bands at 2000, 1897 and 1873 cm^−1^ (Figure 6). The separated ν_2_ and ν_3_ modes and the moderate shift to a smaller wavenumber indicate the presence of the intact Re−Br bond. These observations can be explained by the cleavage of the remote N−H bond in the singly reduced [complex−Br]^•−^, resulting in [complex′−Br]^−^ containing negatively charged NHC−1. This assignment receives strong support from DFT calculations. By applying a scaling factor of 0.988, the calculated ν(CO) wavenumbers of parent [complex−Br] at 2029, 1916 and 1886 cm^−1^ for Re−NHC−1−3 are red−shifted to 2009, 1885 and 1860 cm^−1^ on transformation to [complex′−Br]^−^, that is, by 20, 31 and 26 cm^−1^, as compared with the experimental shifts of 17, 26 and 19 cm^−1^. Based on the (ν(CO) values calculated for the [complex−Br]^•−^ (1971, 1856 and 1840 cm^−1^), this primary reduced product is expected to absorb at about 1960, 1865 and 1850 cm^−1^. The cleavage of the N−H bond is, however, too rapid to allow for the detection of [complex−Br]^•−^ with IR SEC under ambient conditions. The subsequent reduction of [complex′−Br]^−^ at R1′ (Table 2) led to the formation of an unassigned mononuclear tricarbonyl complex absorbing at 1971 and 1857 br cm^−1^. These wavenumbers are much larger than those found for the two−electron reduced [complex−5c]^−^ (ν(CO) at 1912, 1811 br cm^−1^) generated at the cathode from Re−NHC−2. We may safely conclude that the two−electron−reduced product formed from [complex′−Br]^−^ at R1′ also has a five−coordinate geometry (indicated by the unresolved ν_2_ and ν_3_ modes), but with only one of the two added electrons residing in the Re(py−NHC) metallacycle. The molecular structure of this [complex′]^−^ should be elucidated from a detailed spectro−electrochemical study combined with DFT calculations, which was beyond the scope of this project.

Finally, the complex Re−NHC−3 (ν(CO) at 2019, 1925 and 1896 cm^−1^) with the more electron−withdrawing pyrene NHC−backbone instead of phenanthrene showed the same reduction path as Re−NHC−2, with reduction products formed at R1 (dimer [{complex}_2_] (ν(CO) at 1984, 1949, 1871 and 1862 cm^−1^) and R2 (5−coordinate anion [complex−5c]^−^ (ν(CO) at 1914 and 1814 br cm^−1^). Monitoring of the final reductions step at R3 (Table 2) revealed the formation of the 5−coordinate dianion [complex−5c]^2−^ (ν(CO) at 1897 and 1799 br cm^−1^). The moderate decrease in the CO−stretching wavenumbers complies with the ultimate reduction being largely localized on the remote pyrene backbone, in line with the arguments based on the analysis of R3 in the cyclic voltammogram of Re−NHC−3 (see Figure 4a,d).

#### 2.3.2. Electrochemical Oxidation

On the oxidative side, Re−NHC−2 and Re−NHC−3 each present an irreversible wave with the peak potential at +0.87 V (see Appendix A). In the case of Re−NHC−1, the peak potential for its irreversible oxidation is only slightly shifted in the cathodic direction by 70 mV. These anodic responses are typical for compounds containing Re(I) tricarbonyl units [55,56,57], including those with NHC−based chelating ligands [38]. The reactivity is usually attributed to a largely metal−centered oxidation (Re(I)→Re(II)) and concomitant CO loss [51]. However, DFT calculations have shown that the HOMO of Re−NHC−3 largely resides on the pyrene backbone, with small contributions coming from the cyclic carbene and the Re center (Appendix A). The nearest occupied Re−Br−based orbitals (HOMO−1 and HOMO−2) lie at 0.45 and 0.53 eV below the HOMO, respectively. Free pyrene shows the quasi−reversible oxidation in acetonitrile at +0.79 V vs. Fc^+^/Fc [52], which indeed further supports the assignment of the HOMO in Re−NHC−3.

### 2.4. Photocatalytic CO_2_ Reduction 

The Rehm–Weller approach is commonly employed as an approximation to determine whether thermodynamic driving forces exist for the various charge−transfer steps in a proposed photocatalytic assembly [58]. As outlined in the following part of this section, we performed photocatalytic experiments with the Re−NHC−1−3 complexes in DMF with added methanol and 1,3−dimethyl−2−phenyl−2,3−dihydro−1*H*−benzo[d]imidazole (BIH) as a sacrificial electron donor (SED). Although the redox potentials of Re−NHC−1−3 were recorded using acetonitrile as the electrolyte solvent (Table 2), the literature suggests that swapping this for DMF has a relatively small effect (≤0.05 V) on the observed reduction potentials for rhenium(I) tricarbonyl complexes [49,59]. Hence, the potential energy diagram of Figure 7 is based on the measured *E*_o_ (onset potential) values for the reduction of the parent complexes, *E*_(A/A_^−^_)_, and the estimated values of the first reduction potential of the excited species, *E**_(A/A_^−^_)_, were calculated using Equation (1),
(1)$$E_{\left(A/A^{-}\right)}^{*}\approx E_{\left(A/A^{-}\right)}+E_{MLCT}$$
where *E*_MLCT_ is the energy (in eV) derived from the high energy onset of the emission spectra of Figure 2b [10]. To estimate the standard reduction potential, *E^0^*_(CO2/CO),_ for the two−electron reduction of CO_2_ to CO under the applied experimental conditions, similar considerations to those adopted in the works of Delcamp et al. [10,29] were applied, where an upper (i.e., most negative) limit for *E^0^*_(CO2/CO)_ was determined by considering the p*K_a_* of the most acidic species in solution (which is likely to be carbonic acid formed due to the presence of residual moisture). The value for *E^0^*_(CO2/CO)_ in DMF has been reported as −0.73–0.0592 p*K_a_* (V vs. Fc^+^/Fc) [60]. Using the relevant calculated value of p*K_a_* = 7.37 for carbonic acid in DMF [61], the value of *E^0^*_(CO2/CO)_ ≥ −1.16 ≈ −1.2 V is suggested for our photocatalytic experimental conditions. The oxidation potential of BIH in DMF has been reported as +0.28 V (vs. SCE) [62], corresponding to −0.12 V vs. Fc^+^/Fc, if *E^0^*_(Fc_^+^_/Fc)_ is taken as +0.4 V vs. SCE [63]. Referring to Figure 7, the estimated values of *E**_(A/A^−^)_ for each of the Re−NHC complexes are clearly lower in energy by at least 0.87 eV (Re−NHC−3) compared to the oxidation potential of BIH, suggesting that in all cases, it is feasible for this SED to reductively quench the excited state of the complex, leading to the formation of [complex−Br]^•−^. Furthermore, the energetic positions of these radical anions are all higher than the envisaged standard CO_2_ reduction potential by approximately 0.7 eV, suggesting the thermodynamic viability of the electron transfer from the reduced complexes to dissolved CO_2_.

While an essential prerequisite, the existence of favorable potential gradients for the charge−transfer steps required in photocatalysis does not necessarily imply that the required chemical interaction will arise between CO_2_ molecules and the reduced Re−NHC complexes. Fortunately, electrochemistry offers a convenient experimental means of generating the reduced forms of the complexes and monitoring their response to the presence of CO_2_. In view of this, the voltammetric characterization of the complexes was repeated in CO_2_−saturated electrolyte solutions (Figure 4b–d). Re−NHC−2 and Re−NHC−3 exhibit some current enhancement already at R1, suggesting the electrocatalytic activity of the two−electron−reduced five−coordinate [complex−5c]^−^ picking up CO_2_ prior to the dimerization by the zero−electron reaction with the parent [complex−Br]. For Re−NHC−1, no catalytic activity is observed at R1 where, instead, the reductive elimination of H is taking place due to the cleavage of the N−H bond in the one−electron reduced [complex−Br]^•−^, as revealed by IR spectroelectrochemistry. 

The observation of some electrocatalytic activity for Re−NHC−2 and Re−NHC−3 already at R1 confirms the rapid dissociation of the bromide ligand upon the initial reduction. This is a requirement to secure the formation of an adduct intermediary between CO_2_ and the coordinatively−unsaturated reduced metal center.

Having estimated the existence of a driving force for photocatalytic CO_2_ reduction using complexes Re−NHC−1−3, and having confirmed their capacity to act as electrocatalysts, photocatalytic experiments were performed (Appendix A). In the event, all three compounds demonstrated the ability to photocatalytically reduce CO_2_ to CO. Re−NHC−2 and, surprisingly, Re−NHC−1 also offered modest TONs of 10 and 11, respectively, after 24 h of irradiation at 355 nm. By contrast, Re−NHC−3 produced only trace amounts of CO during 24 h at this irradiation wavelength. The situation was reversed, however, when the light source was changed to 470 nm. Here, Re−NHC−3 catalyzed CO production with a TON of 26 over 24 h, while Re−NHC−1 and Re−NHC−2 yielded only trace amounts of CO.

Control experiments were carried out, where the solutions were prepared in the same way but left in the dark without irradiation. No CO production was observed, clearly implying that these catalysts require light to mediate the reduction of CO_2_ to CO. Samples were also prepared containing all components of the photocatalytic system, apart from the catalyst. Under irradiation, this arrangement was also unable to produce CO. Given that the voltammetric and spectroelectrochemical data of Section 2.3.1 confirm the rapid dissociation of [complex−Br]^•−^, thereby implying the possibility of CO_2_ adduct formation, it is tentatively suggested that the well−established photocatalytic CO_2_ reduction mechanism outlined by Ishitani et al. may be applicable to the Re−NHC catalysts described herein [64].

The longer excited−state lifetimes of Re−NHC−2 (Table 1) might normally be predicted to facilitate more efficient CO_2_ reduction due to the extended time frame over which it can accept an electron from the sacrificial agent and undergo quenching to form the reduced (pre−)catalyst species. However, this expectation was not borne out experimentally. In fact, amongst the three complexes, it appears that catalyst stability is the main determinant of CO TON. As discussed above, with respect to photophysical measurements and electrochemical characterization, Re−NHC−3 was observed to be the most stable of the studied series. Its superior resistance to photolytic or solvent−driven breakdown implies that a greater relative concentration of this complex will remain intact in solution and be available for photoexcitation in the initial step of the catalytic cycle. The wavelength variance of the photocatalytic performance of the compounds is consistent with the spectral emission data in Figure 2b, which suggests that a lower energy/ longer wavelength ^3^MLCT transition prevails for Re−NHC−3, compared to the other complexes. As revealed by TD−DFT calculations, the lowest−energy HOMO→LUMO electronic excitation in Re−NHC−3 has a mixed π→π*(NHC−pyrene) and *d*_π_(Re)→π*(pyridine) (IL/MLCT) character. 

The observed photocatalytic performance of 26 TONs of CO offered by Re−NHC−3 is remarkably similar to a plateau TON of approximately 25 reported for a complex similar to that of Figure 1a, except for the replacement of the imidazole portion of the ligand with benzimidazole [29]. In that case, a solar simulator was used as the irradiation source.

Based on the photophysical and electrochemical experiments, it can be concluded that Re−NHC−3, the best performing photocatalyst, is the most (photo)stable of the three complexes.

## 3. Materials and Methods

All chemicals and anhydrous solvents were supplied by Aldrich Chemicals and used under a nitrogen atmosphere. FTIR measurements were carried out using a Perkin−Elmer 2000 FTIR spectrometer. UV−Vis absorption/emission spectra were recorded on a Horiba Scientific Duetta spectrophotometer/spectrofluorimeter equipped with EZSpec software (Kyoto, Japan). All excitation and emission spectra and time−correlated single photon counting (TCSPC) lifetimes were carried out using an Edinburgh Instruments FLS1000 photoluminescence spectrometer (Edinburgh, Scotland). For steady−state measurements, an Xe Arc lamp and a visible PMT−900 detector were used. For lifetimes, a 375 nm variable pulse length diode laser (VPL−375) was employed. All data analyses were carried out using Floracle ® software version 2.15.2. CO TONs were quantified through headspace sampling using a Shimadzu GC−2010 Plus gas chromatograph equipped with LabSolutions Lite 5.5 software (Kyoto, Japan). All NMR spectra (Appendix A) were recorded on a Bruker Avance Ultrashield 600 spectrometer with TopSpin 3.6.1 software and were referenced to the deuterated solvent peak as an internal reference. Elemental analyses were carried out at London Metropolitan University.

### 3.1. Syntheses

#### 3.1.1. 1−(Pyridin−2−yl)−1H−phenanthro [9,10−d]imidazole (NHC−1)

A mixture of 1*H*−phenanthro[9,10−*d*]imidazole (0.33 g, 1.53 mmol), 2−fluoropyridine (0.53 g, 5.51 mmol), K_2_CO_3_ (0.423 g, 3.06 mmol) and KF (0.18 g, 3.06 mmol) in DMF (15 mL) was heated at 130 °C for 3 days. All volatiles were distilled off. The residue was sonicated with diethyl ether (10 mL, twice). The white solid was filtered off and extracted with dichloromethane (20 mL). The solvent was evaporated to dryness in vacuo. Yield: 180 mg (61%).

^1^H NMR (600 MHz, DMSO−*d*_6_, 298 K): δ 8.91 (d, *J*_HH_ = 8.4 Hz, 1H), 8.87 (d, *J*_HH_ = 8.2 Hz, 1H), 8.78 (dd, *J*_HH_ = 5.0 Hz, 1H), 8.63 (dd, *J*_HH_ = 8.1 Hz, 1H), 8.51 (s, 1H), 8.24–8.21 (td, *J*_HH_ = 7.7 Hz, 1.9 Hz, 1H), 7.88 (d, *J*_HH_ = 7.9 Hz, 1H), 7.79–7.72 (m, 2H), 7.71–7.66 (m, 1H), 7.62–7.57 (m, 1H), 7.46–7.41 (m, 1H), 7.35 (dd, *J*_HH_ = 8.4 Hz, 1H).

^13^C {^1^H} NMR (126 MHz, DMSO−*d*_6_, 298 K): δ 150.11, 149.81, 142.82, 140.12, 137.59,

128.48, 127.88, 127.55, 126.83, 126.68, 125.90, 125.57, 125.35, 125.09, 124.45, 123.72,

122.19, 122.11, 121.61, 121.46.



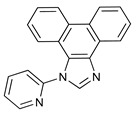



#### 3.1.2. 1−Benzyl−3−(pyridine−2−yl)−1H−3λ^4^−phenanthro[9,10−d]imidazolium bromide (NHC−2)

A mixture of NHC−1 (0.17 g, 0.57 mmol) and excess benzyl bromide (1 mL) were added to a pressure tube, heated at 120 °C for 1–1.5 h and allowed to cool to room temperature. The resulting white solid was washed twice with diethyl ether and twice with ethanol. A pipette was used to remove the solvent. The solid was left in the pressure tube each time. The product was finally dissolved in DCM and transferred to a round bottom flask. The solvent was then removed on a rotary evaporator until dry. Yield: 0.20 g (75.5%).

^1^H NMR (600 MHz, DMSO−*d*_6_, 298 K): δ 10.34 (s, 1H), 9.09 (t, *J*_HH_ = 7.7 Hz, 2H), 8.93 (dd, *J*_HH_ = 5.0 Hz, 1H), 8.46–8.42 (m, 2H), 8.24 (d, *J*_HH_ = 8.0 Hz, 1H), 8.02–7.98 (m, 1H), 7.87–7.81 (m, 2H), 7.75 (t, *J*_HH_ = 7.6 Hz, 1H), 7.59 (t, *J*_HH_ = 7.9 Hz, 1H), 7.50 (d, *J*_HH_ = 7.5 Hz 2H), 7.44 (t, *J*_HH_ = 7.5 Hz, 2H), 7.39–7.36 (m, 1H), 7.15 (dd, *J*_HH_ = 8.6 Hz, 1H), 6.37 (s, 2H).

^13^C {^1^H} NMR (600 MHz, DMSO−*d*_6_, 298 K): δ 150.5, 147.6, 142.8 (N−C*H*−N), 141.1, 133.5, 129.7, 129.5, 129.2, 128.5, 128.46, 124.4, 128.2, 127.5, 126.9, 126.5, 125.6, 124.9, 124.7, 122.8, 122.7, 121.6, 119.8, 119.7, 53.3.



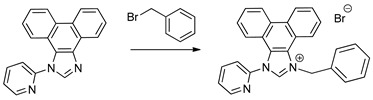



#### 3.1.3. 1−(Pyridin−2−yl)−3a^1^, 5a^1^ −dihydro−9H−11λ^4^ −pyreno[4,5−d]imidazole

Potassium carbonate (2.74 g, 19.8 mmol) and potassium iodide (3.29 g, 19.8 mmol) were transferred to a 100−mL Schlenk tube containing 9*H*−pyreno[4,5−*d*]imidazole (3.20 g, 13.20 mmol), and the contents were then dried under high vacuum for 1 h. Dry dimethyl sulfoxide (DMSO, 20 mL) was then added under an inert atmosphere. The mixture was stirred at room temperature until all the material had completely dissolved. Then, 2−fluoropyridine (ca. 1.5 mL) was added dropwise, the condenser was fitted upright, and the reflux commenced.

The reaction was run for 3 days at 120 °C, affording a dark brown solution. DMSO was then distilled off under reduced pressure via vacuum distillation, having yielded a brown−colored solid. The solid was then washed with diethyl ether to remove unwanted by−products. The product was extracted from the solid matter with DCM and isolated under reduced pressure using a rotary evaporator. Yield: 2.5 g (59.3%).

^1^H NMR (600 MHz, DMSO−*d*_6_) δ 8.88 (dd, *J*_HH_ = 7.6, 1.2 Hz, 1H), 8.84 (ddd, *J*_HH_ = 4.8, 1.9, 0.8 Hz, 1H), 8.65 (s, 1H), 8.35–8.25 (m, 2H), 8.25–8.21 (m, 2H), 8.21–8.15 (m, 2H), 7.98 (dt, *J*_HH_ = 8.0, 1.0 Hz, 1H), 7.87 (t, *J*_HH_ = 7.8 Hz, 1H), 7.80 (ddd, *J*_HH_ = 7.6, 4.9, 1.0 Hz, 1H), 7.64 (dd, *J*_HH_ = 8.0, 1.1 Hz, 1H).

^13^ C {^1^ H} NMR (126 MHz, DMSO−*d*_6_, 298 K): δ 150.6, 150.3, 143.6, 140.6, 140.1, 138.6, 132.2, 131.8, 128.3, 128.2, 127.1, 126.5, 126.4, 126.2, 125.6, 125.2, 125.1, 122.9, 122.1, 121.9, 119.7, 119.3.



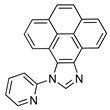



#### 3.1.4. (9−Benzyl−11−(pyridin−2−yl)−3a^1^, 5a^1^ −dihydro−9H−11λ^4^ −pyreno[4,5−d]imidazolium bromide (NHC−3)

A mixture of 1−(pyridin−2−yl)−3a^1^, 5a^1^ −dihydro−9H−11λ^4^ −pyreno[4,5−*d*]imidazole

0.20 g, 0.63 mmol) and an excess of benzyl bromide (1 mL) were added to a pressure tube, heated at 120 °C for 1–1.5 h and allowed to cool to room temperature. The resulting solid was washed twice with diethyl ether and twice with ethanol. A pipette was used to remove the solvent. The solid was left in the pressure tube each time and finally dissolved in DCM. The solution was then transferred to a round bottom flask and the solvent was removed on a rotary evaporator until dry, to obtain a white powder. Yield: 0.180 g (58.6%).

^1^H NMR (600 MHz, DMSO−*d*_6_) δ 10.49 (s, 1H), 9.02–8.97 (m, 1H), 8.70 (dd, *J*_HH_ = 8.0, 1.0 Hz, 1H), 8.50 (td, *J*_HH_ = 7.8, 1.9 Hz, 1H), 8.49–8.44 (m, 1H), 8.42 (dd, *J*_HH_ = 7.9, 1.0 Hz, 1H), 8.33 (dd, *J*_HH_ = 7.9, 1.0 Hz, 1H), 8.30–8.24 (m, 2H), 8.12 (t, J = 7.9 Hz, 1H), 8.06 (ddd, *J*_HH_ = 7.7, 4.9, 1.0 Hz, 1H), 7.97 (t, *J*_HH_ = 7.9 Hz, 1H), 7.61–7.57 (m, 2H), 7.46 (dd, *J*_HH_ = 8.5, 7.0 Hz, 2H), 7.43–7.35 (m, 2H), 6.50 (s, 2H).

^13^C {^1^H} NMR (151 MHz, DMSO−*d*_6_) δ 151.0, 148.1, 143.4, 141.7, 134.0, 132.1, 131.9, 129.7, 129.1, 128.76, 128.6, 128.2, 128.1, 128.06, 127.6, 127.5, 127.3, 126.7, 123.5, 123.4, 123.3, 121.3, 119.9, 119.4, 53.8. 



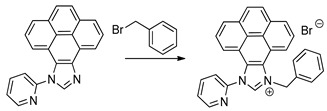



#### 3.1.5. Re−NHC−1

Solid NHC−1 (0.10 g, 0.34 mmol) and [ReCO_5_Br] (0.21 g, 0.51 mmol) were added to toluene (10 mL) with triethylamine (1 mL), purged beforehand with N_2_ for 20 min. The mixture was refluxed overnight at 110 °C. A yellow precipitate was formed. The solvent was decanted off and the precipitate was washed multiple times with pentane and diethyl ether. Excess solvent was removed under reduced pressure to leave behind a bright yellow powder. Yield: 0.16 g (48.5%).

^1^H NMR (600 MHz, DMSO−*d*_6_, 298 K): δ 14.82 (s, 1H), 9.06 (d, *J*_HH_ = 7.9 Hz, 1H), 9.04–9.01 (m, 2H), 8.80 (d, *J*_HH_ = 7.9 Hz, 1H), 8.37–8.32 (m, 2H), 8.28 (d, *J*_HH_ = 8.3 Hz, 1H), 7.92 (t, *J*_HH_ = 7.5 Hz, 1H), 7.86 (t, *J*_HH_ = 7.9 Hz, 1H), 7.81 (t, *J*_HH_ = 7.6 Hz, 1H), 7.76 (t, *J*_HH_ = 7.9 Hz, 1H), 7.58 (t, *J*_HH_ = 6.5 Hz, 1H).

^13^C {^1^H} NMR (126 MHz, DMSO−*d*_6_, 298 K): δ 197.55, 197.15, 195.75, 189.28, 154.07, 153.67, 141.68, 130.91, 128.94, 128.67, 128.18, 127.66, 126.87, 126.77, 124.96, 124.35, 124.06, 123.92, 123.58, 122.31, 120.40, 120.07, 115.93.

IR spectroscopy (THF, ν(CO), cm^−1^): 2017, 1924, 1892

Mass spectrometry (Maldi+): 644.96 (M^+^)

Anal. Calcd (%) for C_23_H_13_BrN_3_CO_3_Re (645.48): C, 42.80; H, 2.03; N, 6.51. Found (%): C, 42.71; H, 2.09; N, 6.44.

#### 3.1.6. Re−NHC−2

The mixture NHC−2 (0.20 g, 0.43 mmol) and [ReCO_5_Br] (0.26 g, 0.64 mmol) was added to THF and toluene (20 mL, 1:1, *v*/*v*), purged beforehand with nitrogen for 20 min. The mixture was refluxed at 90 °C overnight. The solvents were then removed on a rotary evaporator. The yellow solid was washed twice with hexane and diethyl ether and dried. The product was purified by column chromatography on neutral alumina, with THF−toluene 1:1 (*v*/*v*) as the mobile phase. The product was then eluted with neat DCM, separated by solvent evaporation, and dried.

Yield: 0.23 g (yield 48.6%).

^1^H NMR (600 MHz, DMSO−*d*_6_, 298 K): δ 9.06 (d, *J*_HH_ = 5.2 Hz, 1H), 8.98 (d, *J*_HH_ = 8.1 Hz, 2H), 8.35 (t, *J*_HH_ = 7.8 Hz, 1H), 8.29 (d, *J*_HH_ = 7.9 Hz, 2H), 8.23 (d, *J*_HH_ = 8.3 Hz, 1H), 7.80 (t, *J*_HH_ = 7.4 Hz, 1H), 7.77–7.69 (m, 2H), 7.66–7.60 (m, 2H), 7.58–7.38 (br s, 2H), 7.35 (t, *J*_HH_ = 7.4 Hz, 2H), 7.31–7.26 (m, 1H), 6.58–6.10 (br, 2H).

^13^ C NMR {^1^H} (126 MHz, DMSO−*d*_6_, 298 K)): δ 201.2, 197.47, 196.94, 188.28, 153.90, 153.73, 141.58, 139.17, 134.85, 129.54, 129.28, 129.00, 128.85, 128.19, 128.09, 127.50, 127.39, 127.01, 126.61, 126.13, 124.97, 124.89 124.44, 124.30, 123.75, 122.12, 120.01, 119.88, 116.76, 55.97.

IR spectroscopy (THF, ν(CO), cm^−1^): 2019, 1927, 1896

Mass spectrometry (Maldi+): *m*/*z* 735.02 (M^+^)

Anal. Calcd (%) for C_30_H_19_BrN_3_CO_3_Re (735.60): C, 48.98; H, 2.60; N, 5.71. Found (%): C, 49.02; H, 2.49; N, 5.76.

#### 3.1.7. Re−NHC−3

The same synthetic procedure as described above for Re−NHC−2 was used, although NHC−3 (0.36 g, 0.74 mmol) and [ReCO_5_Br] (0.45 g, 1.11 mmol) were used instead.

Yield: 0.455 g (54.4%).

^1^H NMR (600 MHz, CDCl_3_, 298 K): δ 9.11 (d, *J*_HH_ = 5.7 Hz, 1H), 8.55 (d, *J*_HH_ = 7.9 Hz, 1H), 8.52 (d, *J*_HH_ = 8.1 Hz, 1H), 8.28 (d, *J*_HH_ = 8.5 Hz, 2H), 8.24 (d, *J*_HH_ = 7.7 Hz, 1H), 8.14–8.12 (m, 2H), 8.12–8.09 (m, 1H), 8.06 (t, *J*_HH_ = 7.9 Hz, 1H), 7.95 (t, *J*_HH_ = 7.9 Hz, 1H), 7.59–7.48 (br, 2H), 7.44 (t, *J*_HH_ = 6.1 Hz, 1H), 7.39 (t, J_HH_ = 7.8 Hz, 2H), 7.32 (t, *J*_HH_ = 7.5 Hz, 1H), 6.61–6.38 (br, 2H).

^13^ C {^1^H} NMR (126 MHz, DMSO−*d*_6_, 298 K)): δ 201.2, 197.5, 196.9, 188.3, 154.0, 154.9, 141.6, 137.3, 134.9, 131.4, 131.1, 129.8, 128.9, 128.2, 127.9, 127.8, 127.8, 127.1, 126.6, 126.5, 126.4, 125.7, 125.3, 124.4, 122.4, 122.2, 121.5, 119.9, 118.9, 118.9, 116.9, 55.6.

IR spectroscopy (THF, ν(CO), cm^−1^): 2019, 1927, 1896

Mass spectrometry (Maldi+): *m*/*z* 782.00 (M + Na)^+^

Anal. Calcd for C_32_H_19_BrN_3_O_3_Re (759.62): C, 50.60; H, 2.52; N, 5.53. Found: C, 50.52; H, 2.49; N, 5.49.

### 3.2. X-ray Crystallography

A crystal of Re−NHC−1 was mounted under Paratone−N oil and flash−cooled to 100 K under nitrogen in an Oxford Cryosystems Cryostream. Single−crystal X-ray intensity data were collected using a Rigaku XtaLAB Synergy diffractometer (Mo Kα radiation (λ = 0.71073 Å)). The data were reduced within the CrysAlisPro software [65]. The structure was solved using the program Superflip [66] and all non−hydrogen atoms located. Least−squares refinement against *F* was carried out using the CRYSTALS suite of programs [67]. The non−hydrogen atoms were refined anisotropically. All the hydrogen atoms were located in difference Fourier maps. The hydrogens attached to carbon were then placed geometrically with a C−H distance of 0.95 Å and a *U*_iso_ of ~1.2 times the value of *U*_eq_ of the parent C atom and refined with riding constraints. The position of the hydrogen attached to nitrogen, N(20), was refined with a N−H distance restraint of 0.85(1) Å and a *U*_iso_ of ~1.2 times the value of *U*e_q_ of the parent N atom. CCDC code: 2244937. The supplementary crystallographic data can be obtained free of charge via www.ccdc.cam.ac.uk/data_request/cif (accessed on 12 February 2023), or by emailing data_request@ccdc.cam.ac.uk, or by contacting The Cambridge Crystallographic Data Centre, 12 Union Road, Cambridge CB2 1EZ, UK; fax: +44 1223 336033.

### 3.3. Photophysical Studies

UV−Vis absorption, emission and excitation spectra (both at ambient temperature and 77 K) were recorded on a Horiba Scientific Duetta spectrophotometer/spectrofluorimeter. The lifetimes were determined using an Edinburgh Instruments FLS 1000 photoluminescence spectrometer. All measurements at ambient temperature were conducted in dry acetonitrile and those at 77 K in a glass of ethanol−methanol 4:1 (*v*/*v*). The excitation wavelength was 365 nm, and the solutions were deaerated using the freeze−pump−thaw method.

### 3.4. Cyclic Voltammetry

Cyclic voltammetric (CV) measurements were performed at room temperature using a CHI 750C Electrochemical Workstation. The electrolytes contained 1 mM complexes in dry acetonitrile/0.1 M tetrabutylammonium hexafluorophosphate (TBAPF_6_). A glassy carbon disk was used as the working electrode, with a platinum wire as the counter electrode. A pseudo−reference electrode was employed, consisting of a silver wire immersed in the supporting electrolyte and separated from the rest of the electrolyte solution by a porous tip. All potentials are reported relative to the standard ferrocenium/ferrocene (Fc^+^/Fc) couple.

The cyclic voltammetric measurements were conducted in septum−sealed V−type glass cells. Prior to each measurement, 2.5 mL of the electrolyte solution was placed in the cell, the electrodes were inserted, and the liquid level was carefully marked on the cell wall. An additional small volume (≤0.5 mL) of dry acetonitrile was injected into the cell before purging it with either N_2_ or CO_2_. Purging was continued until the electrolyte volume reduced back to the 2.5 mL mark in ca. 20 min. A blanket of either N_2_ or CO_2_ was maintained above the electrolyte solution during the measurements.

### 3.5. IR Spectroelectrochemistry

IR spectroelectrochemical experiments were performed using an air−tight OTTLE cell (Spectroelectrochemistry Reading, UK) [68] positioned in the sample compartment of a Bruker Vertex 70v FT−IR spectrometer equipped with a DLaTGS detector. The cell was equipped with Pt minigrid (32 wires per cm) working and auxiliary electrodes, an Ag−microwire pseudo−reference electrode and optically transparent CaF_2_ windows. The course of each of the spectroelectrochemical experiments was monitored by thin−layer cyclic voltammetry; the potential control was realized with an EmStat3+ potentiostat (PalmSens, The Netherlands) operated with the PSTrace5 software. The concentration of the spectroelectrochemical samples was ca. 2 × 10^−3^ mol dm^−3^. Dry 10^−1^ M TBAPF_6_ was used as the supporting electrolyte.

### 3.6. Photocatalytic CO_2_ Reduction

The samples contained a 0.13 mM catalyst and 10 mM sacrificial agent (BIH) in DMF−methanol 6:2 (*v*/*v*). They were prepared in an 18−mL Schlenk vessel, deaerated by 3 freeze−pump−thaw cycles, purged by CO_2_ for 20 min and irradiated for 24 h using a 355−nm or 470−nm LED. In addition, 1 mL of headspace was collected and injected into a Shimadzu GC−2010 Plus gas chromatograph. A 10,000−ppm internal standard (CO) was used to calibrate the TON and TOF values.

### 3.7. Computational Details

The electronic structures were calculated by density functional theory (DFT) methods using the Gaussian 16 quantum chemical package [69]. The calculations employed B3LYP [70,71] hybrid functional (G16/B3LYP). Polarized double−ζ basis sets 6−31 + G(d) were used for H, C, O and N atoms, and triple−ζ basis sets 6−311 + G(d) containing the polarization function were used for the Br atom. For the Re atom, we used a large basis set (8s7p6d2f1g)/[6s5p3d2f1g] and a standard Stuttgart/Dresden pseudopotential for 60 core electrons [72,73]. Open−shell systems were calculated by the unrestricted Kohn–Sham approach (UKS). Geometry optimization followed by vibrational analysis was made in vacuum (no imaginary frequency). Solvent effects (acetonitrile) were described by the polarizable conductor calculation model (CPCM) [74,75].

## 4. Conclusions

The syntheses and characterization have been reported for three novel *N*−heterocyclic carbene rhenium tricarbonyl complexes, Re−NHC−1−3, acting as photocatalysts for CO_2_ conversion to CO. Compared to similar [Re]−NHC ([Re] = Re(CO)_3_(halide)) complexes previously evaluated for photocatalytic and electrocatalytic CO_2_ reduction, the studied series offers the possibility of extending the π electron conjugation of the NHC ring by fusing either phenanthrene or pyrene backbones to the imidazole (NHC) part of the ligand. While the emission spectra of the phenanthrene−containing complexes, Re−NHC−1 and Re−NHC−2, were comparable to those reported for other [Re]−NHC species, the emission spectra for the pyrene−containing molecule, Re−NHC−3, are substantially red−shifted, suggesting a lower−energy ^3^IL/MLCT transition than is commonly observed for this class of compounds. The replacement of imidazole N−H (Re−NHC−1) with N−benzyl (Re−NHC−2) was found to confer greater photostability. The extended π−conjugation within the chelating ligand also contributed to stability, with Re−NHC−3 being the least prone to Re−Br photosubstitution in acetonitrile.

The electrochemical characterization of the studied complexes revealed the thermodynamic driving force for photocatalytic CO_2_ reduction. With the aid of cyclic voltammetry and IR spectroelectrochemical measurements, the first reduction wave has been attributed to a two−electron process in the cases of Re−NHC−2 and Re−NHC−3. A single−electron process prevails for Re−NHC−1, which undergoes the reductive cleavage of the imidazole N−H bond. For each complex, the two−electron reduction triggered catalytic current enhancement upon the CO_2_ saturation of the electrolyte solution. Encouraged by these results, photocatalytic experiments confirmed the ability of all three compounds to reduce CO_2_ to CO. The most promising performance was observed for Re−NHC−3, which produced 26 TONs of CO when irradiated at 470 nm, i.e., approaching the green region of the visible light spectrum. The other two complexes were inactive at this wavelength and offered poorer performance even under higher−energy irradiation. The superior performance of Re−NHC−3 is attributable to the extended conjugation offered by the presence of the backbone pyrene group. This moiety serves to reduce electron density at the metal center which, in turn, inhibits photo−induced halide ligand loss from the neutral complex, thereby increasing its stability. The longer−wavelength photocatalytic performance complies with the characteristic lower energy of the ^3^IL/MLCT transition induced by the incorporation of the extended pyrene aromatic system.

In summary, we have demonstrated how a rational ligand design can be used to extend the light−harvesting capacity of [Re]−NHC tricarbonyl compounds towards the red end of the visible spectral range, while preserving their simultaneous function as catalytic centers for CO_2_ reduction.

## Data Availability

Not applicable.

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
