# Peer review of "Ligand−Structure Effects on N−Heterocyclic Carbene Rhenium Photo− and Electrocatalysts of CO2 Reduction"

_molecules, 2023, doi:10.3390/molecules28104149_

Round 1

Reviewer 1 Report

See attached file

Reviewer 2 Report

In this manuscript, Kearney et al. described “Ligand-structure effects on N-heterocyclic carbene rhenium photo- and electrocatalysts of CO2 reductio” in detail. Samples are properly characterized, and the activities are excellent. However, at this stage there are still many problems and I therefore suggest a major review for this manuscript keeping in mind the following questions.

1) What is meant by “Electronic Absorption”? Should it not be light absorption?

2) The introduction is meaningless and needs proper attention. The respected authors are requested to write a precise introduction.

3) Why does the introduction of N-heterocyclic carbene induces CO2 conversion enhancement?  The respected authors are requested to write a precise mechanism and related references for the mentioned statement.

4) A detailed schematic diagram showing the charge separation and electron donating/attracting capacity of the ligand/acceptor for the reduction of CO2 must be provided to check the redox reaction pathways in a glance.

5) No specific experiment has been conducted for charge separation which plays major role in the overall redox reactions. How does the author claim that charge has been transferred for the reduction of CO2?

6) What is the effect of the ligand on the thermodynamic energy of the electrons as the thermodynamic energy of the electrons plays crucial role in reduction process. Higher the thermodynamic energy of the electrons, fast is the kinetics of CO2 reduction.

7) Some important citations are missing.

i)  A. Zada, N. Ali, F. Subhan, N. Anwar, M. I. A. Shah, M. Ateeq, Z. Hussain, K. Zaman, M. Khan, Suitable energy platform significantly improves charge separation of g-C3N4 for CO2 reduction and pollutant oxidation under visible-light, Prog. Nat. Sci. Mat. Int. 29 (2019) 138-144.

ii) F. Raziq, K. Khan, S. Ali, S. Ali, H. Xu, I. Ali, A. Zada, P. M. Ismail, A. Ali, H. Khan, X. Wu, Q. Kong, M. Zahoor, H. Xiao, X. Zu, S. Li, L. Qiao, Accelerating CO2 reduction on novel double perovskite oxide with sulfur, carbon incorporation: Synergistic electronic and chemical engineering, Chem. Eng. J. 446 (2022) 137161.

8) What is the difference between photo-reduction and photocatalytic reduction? The respected authors must explain in detail so that the reader may know the actual difference and do not mix the two different terminologies.

9) What are the major reduction products formed during the photo- and electrocatalysts of CO2 and what are the factors that affect their production in both processes?

10) There are many semiconductor materials, why have the authors chosen the mentioned photo- and electrocatalysts for CO2 reduction and what is the novelty as many authors used the same materials for CO2 reduction.

11) What is the difference between “Photoreactivity” and photocatalytic reactivity? The respected authors must explain in detail so that the reader may know the actual difference and do not mix the two different terminologies.

Moderate errors

Reviewer 3 Report

This paper should not be published for the lack of both novelty and practicability.

First, the reaction is old and there has been a lot of example reported, including the Re-carbene complex catalysis (e.g. 10.1039/c9dt02533b, acs.energyfuels.1c02372, etc). This subject has been well investigated and this work does not provide new information.

Second, Re is a rare metal, using this metal for the catalytic reaction to produce the abundant product itself might be a strange idea, especially in the occasion that a lot of references have reported similar reactions catalyzed by cheap metals (e.g. 10.1021/acs.organomet.8b00535, using manganese N-heterocyclic carbene).

Third, the paper is not well prepared. It not only fails to express what’s new in this work, but also fails to clarify the structure-activity relationship of the materials. Key characterizations such as XPS are missed. These characterizations of the catalyst before and after reaction are important because they can provide the information about the valent changes of the catalytic metals. Moreover, the status of the catalyst during the reaction should also be carefully investigated. The present mechanism discussion is insufficient and inconvincible, because key control experiments are missed. For example, the intermediate of the catalytic species should be captured by techniques.

Overall, the paper is not well written as well. The language should be polished. The graphics are not prepared professionally. These drawbacks indicate that this is not a professional paper for publication.

English should be improved.

Round 2

Reviewer 2 Report

The authors made significant changes to improve the quality of the paper, I therefore accept the publication of this paper in your reputed journal.

Reviewer 3 Report

Still unacceptable

ok